# Temperature Field Reconstruction Method for Acoustic Tomography Based on Multi-Dictionary Learning

**DOI:** 10.3390/s23010208

**Published:** 2022-12-25

**Authors:** Yuankun Wei, Hua Yan, Yinggang Zhou

**Affiliations:** School of Information Science and Engineering, Shenyang University of Technology, Shenyang 110870, China

**Keywords:** acoustic tomography, temperature field reconstruction, multi-dictionary learning, peak-type classifier, sparse representation, OMP

## Abstract

A reconstruction algorithm is proposed, based on multi-dictionary learning (MDL), to improve the reconstruction quality of acoustic tomography for complex temperature fields. Its aim is to improve the under-determination of the inverse problem by the sparse representation of the sound slowness signal (i.e., reciprocal of sound velocity). In the MDL algorithm, the K-SVD dictionary learning algorithm is used to construct corresponding sparse dictionaries for sound slowness signals of different types of temperature fields; the KNN peak-type classifier is employed for the joint use of multiple dictionaries; the orthogonal matching pursuit (OMP) algorithm is used to obtain the sparse representation of sound slowness signal in the sparse domain; then, the temperature distribution is obtained by using the relationship between sound slowness and temperature. Simulation and actual temperature distribution reconstruction experiments show that the MDL algorithm has smaller reconstruction errors and provides more accurate information about the temperature field, compared with the compressed sensing and improved orthogonal matching pursuit (CS-IMOMP) algorithm, which is an algorithm based on compressed sensing and improved orthogonal matching pursuit (in the CS-IMOMP, DFT dictionary is used), the least square algorithm (LSA) and the simultaneous iterative reconstruction technique (SIRT).

## 1. Introduction

Obtaining accurate temperature distribution information is of great significance in many fields. The combustion condition monitoring of the power station boiler is highly critical to ensure the stable operation of the system and improve the combustion efficiency, where the change in temperature distribution in the furnace directly reflects the actual combustion situation of the furnace. Therefore, in order to achieve optimal combustion, as well as safe and efficient operation of the boiler, it is necessary to conduct real-time and accurate monitoring of the temperature distribution inside the boiler [1,2,3,4,5]. The change of atmosphere and ocean has an important impact on the climate and human survival activities, and these changes are also reflected in the change of temperature distribution to a certain extent [6,7,8,9]. Therefore, the monitoring of temperature distribution in the atmosphere and ocean is not only of great practical significance, but also has broad prospects for development. When the grain is attacked by pests, mildew and abnormal water, the temperature of some parts of the grain will rise, called hot spots [10]. In such cases, the monitoring of grain temperature distribution can not only ensure the safe storage of grain and reduce grain loss, but also have good social and economic benefits.

The measurement methods for temperature distribution are mainly divided into contact type and non-contact type. Among these, contact type methods (such as thermocouples) are subject to many limitations when used in the above-mentioned application scenarios. For example, while measuring a high-temperature furnace, the contact temperature sensor may be damaged due to the high-temperature environment. In addition, the contact measurement method requires a large number of sensors (such as thermocouples) for the grain to compensate for the low thermal diffusivity of the grain, making the system maintenance difficult. On the other hand, non-contact methods include laser-based, radiation-based and acoustic methods. Laser-based measuring systems are very expensive and difficult to implement for large-scale boilers, while the radiation-based methods give inaccurate results in coal-fired boilers because ash often smears the camera lenses. In contrast, the technology of reconstructing the temperature field using an acoustic CT (AT) could overcome the above shortcomings to some extent, thus providing relatively high accuracy.

AT temperature field reconstruction technology calculates the distribution of sound slowness (i.e., the reciprocal of sound speed) in the measured area by measuring the time of flight (TOF) of sound waves passing through the measured area in multiple directions. The temperature distribution is then obtained using the relationship between sound speed and temperature. When AT technology is employed to measure the temperature field, it is necessary to place multiple acoustic transceivers around the measured area to form *M* direct acoustic paths through the measured area and divide the area into *N* grids. After the sound wave TOF on each direct path is measured, a temperature value can be reconstructed for each grid by using an appropriate reconstruction algorithm, assigning them in the grid center. Given the limited availability of TOF data, the reconstruction algorithm with *N* greater than *M* is more suitable for the reconstruction of a complex temperature field. However, when *N* is greater than *M*, the inverse problem is under-determined. In general, the larger the *N* value, the more ill-conditioned the inverse problem is; that is, the more sensitive the reconstruction result is to the TOF measurement error.

Most reconstruction algorithms assume uniform temperature distribution in the grid, and the mathematical model is established based on the length of sound paths in the grid. Typical reconstruction algorithms include least squares algorithm (LSA) [2,11], synchronous iterative reconstruction technology (SIRT) [12], etc. These approaches require *M* > *N* to avoid under-determination of the inverse problem. Therefore, it is difficult to provide complete and detailed temperature distribution information [11].

To address this issue, many scholars have proposed various two-stage methods. Zhong et al., in the first stage of one such two-stage method, expressed the temperature field reconstruction model as the least squares problem and solved the temperature distribution under a coarse grid with the improved fast iterative shrinkage thresholding algorithm (FISTA). Progressively, in the second stage, the kernel extreme learning machine (KELM) was used to obtain the temperature distribution under the fine grid. By introducing a positive definite weight matrix, the number of iterations of FISTA was reduced, thereby lowering the time costs of the algorithm. Owing to strong learning ability of KELM, this algorithm has a lower reconstruction error, compared with the traditional algorithm [13]. Jia et al. [11] used the LSA method to reconstruct the temperature field on the coarse grid, and then used the radial basis function approximation model to obtain the temperature distribution on the fine grid. Wu et al. adopted a similar method [11], and used a population-based heuristic random search algorithm (differential evolution algorithm) to optimize radial basis function (RBF) and shape parameters to minimize root mean square error [2]. Fundamentally, the two-stage method is not a fixed algorithm, but a framework for solving inverse problems. Likewise, this idea of two-stage methods was also applied in the ultrasonic tomography where, in the first stage of rough grid re-construction, Zhong et al. transformed the temperature field reconstruction problem into an optimization problem, and then solved the optimization problem with the equilibrium optimizer (EO). Notably, for a complex temperature distribution, EO may fall into the local optimal solution; to solve this problem, they proposed a non-linear time parameter strategy and a new population update rule to improve the EO. Afterwards, in the second stage, Gaussian process expression was used to reconstruct the temperature distribution under the fine grid [14].

Different from general algorithms, some scholars proposed specific reconstruction algorithms. Pal et al. proposed a temperature field reconstruction algorithm that is specially used for the high-temperature area concentrated in the central area of the temperature measurement plane. This algorithm is mainly applied to the surface layer above the blast furnace burden. The algorithm adopts the sampling method of spatial movement. More acoustic paths were able to be involved in the calculation of the slowness function of each grid cell, through shifting the distribution of grid cells [15]. Dokhanchi et al. proposed a method to optimize the layout of acoustic transceivers. By optimizing the coverage of sound paths and maximizing differentiation of TOF value, AT temperature field reconstruction is more suitable for indoor use [6]. In order to reduce the online reconstruction time, Bao et al. proposed two reconstruction algorithms. One is to separate the iterative process from the online reconstruction and execute it offline [12]. The other algorithm is based on online time-resolved, which reduces the measurement time of each frame [16].

Existing reconstruction algorithms typically only consider measurement information. Li Yanqiu et al. proposed an objective function that also takes into account acoustic measurement information, spatial constraints of temperature field and dynamic development information, and solved the objective function by combining Tikhonov regularization and optimization [17].

The *N* > *M* reconstruction algorithm has been proposed in some literature, and compressed sensing theory, or regularization technology, has been used to address the seriously ill-posed inverse problem. Yan et al. proposed an algorithm based on compressed sensing and improved orthogonal matching pursuit (CS-IMOMP). Using the sparsity of the acoustic slowness signal in the discrete Fourier transform (DFT) domain, this approach transforms the reconstruction of the acoustic slowness signal into the reconstruction of sparse signal with few non-zero elements. In addition, the column selection strategy of the OMP algorithm was also improved to reduce the time consumption of temperature field reconstruction [18]. Zhou et al. used the total variation method to solve the ill-conditioned inverse problem. A small constant was introduced to make the total variation function differentiable near the origin, and the Newton–Raphson iteration method was used to solve the objective function, based on total variation minimization [19].

Unlike the above-mentioned algorithm, some scholars did not assume that the sound slowness (temperature) distribution in each grid was uniform when developing the mathematical model. Instead, they used a linear combination of RBFs to estimate the sound slowness distribution; i.e., Markov RBF [10,20], inverse multiquadrics RBF [21,22], multiquadric RBF [23], reflected sigmoidal RBF [24], hybrid RBF of exponential kernel and cubic kernel [25] and Gaussian RBF [5]. This kind of algorithm allows *N* > *M*; in order to address seriously ill-conditioned inverse problems, these algorithms employed methods such as singular value decomposition (SVD), Tikhonov regularization and so on.

Each of the above algorithms has its own application limitations, and the research on the AT temperature field reconstruction algorithm is an open problem.

According to the theory of sparse representation [26], natural signals can be represented as sparse signals in some dictionaries (also called sparse basis). In a sparse signal vector or matrix, the vast majority of elements are zero or very close to zero. By taking signal sparsity as the constraint condition, the range of solutions [27,28] and the difficulty of signal reconstruction can be reduced. There exist two main categories of the algorithms to reconstruct a sparse signal: the greedy algorithm and convex optimization. The greedy algorithm does not seek the global optimal solution, but selects the local optimal solution in each step. Therefore, the greedy algorithm can generally return a satisfactory solution quickly. The typical greedy algorithms include the OMP algorithm and its various improved variants, such as gOMP and SAMP, respectively, etc. In contrast to greedy algorithms, convex optimization seeks the global optimal solution, and thus has a higher reconstruction accuracy. However, the time consumption of such an algorithm is significantly higher than that of the greedy algorithm. Common convex optimization algorithms include basis pursuit, FISTA, etc. It is usually necessary to find a suitable trade-off between the time cost and accuracy, when selecting a sparse reconstruction algorithm.

In this paper, the signal sparse representation was employed to deal with the under-determination of the inverse problem, and the temperature field reconstruction in which the number of grids was greater than the number of acoustic paths was realized. As the reciprocal of sound velocity, which cannot be infinite, no zero elements are present in the sound slowness signal vector. However, the sound slowness (temperature) distribution can be reconstructed by using a suitable dictionary to express the sound slowness signal as a sparse signal, and then be reconstructed from the sparse signal. There are two kinds of dictionary construction methods, namely, the analytic method and the learning method. The analytic method uses predefined mathematical transformations, such as DFT and discrete cosine transform (DCT), to construct a dictionary. The learning method uses a sample set and dictionary learning algorithm to construct the dictionary. The analytic method is widely used because of its simplicity and ease of implementation, but its signal expression form is single and does not have adaptivity. Compared with the analytic method, the dictionary atoms (i.e., column vectors) constructed by the learning method are more abundant in morphology and can better match the structure of the signal or image itself [26]. Therefore, this paper proposes an AT temperature field reconstruction approach based on multi-dictionary learning, called the MDL algorithm. In order to obtain better sparse effect, the MDL uses a learning method to construct dictionaries for temperature (sound slowness) distribution of different peak-types, and uses a peak-type classifier to jointly use multiple dictionaries. Simulation and actual temperature distribution reconstruction experiments show that, compared with LSA, SIRT and CS-IMOMP, the proposed MDL algorithm has better reconstruction quality.

The remainder of this paper is organized as follows: The MDL algorithm is introduced in Section 2; Section 3 conducts numerical simulation and experimental verification of the MDL algorithm; Finally, the conclusions of this paper are given in Section 3.

## 2. MDL Algorithm

### 2.1. Algorithm Overview

The relationship between the propagation velocity *c* (m/s) of sound waves in a gas medium and the absolute temperature *T* (K) of a gas medium can be expressed as [1,10]:(1)$$c=z\sqrt{T},$$
where *z* is the sound constant that is determined by the composition of the gas medium. For air, the value is 20.05. Temperature can be measured using the relationship between sound velocity (or sound slowness) and temperature.

Assuming that the measured area is a two-dimensional plane, the temperature distribution on it is represented by *T*(*x*,*y*), and the sound slowness distribution by *f*(*x*,*y*). Τhe TOF along the *j*th path *p_j_* can be represented as [10]:(2)$$t_{j}=\int_{p_{j}}^{}\frac{dl}{Z\sqrt{T(x,y)}}=\int_{p_{j}}^{}f(x,y)dl,j=1,2,\ldots ,M,$$

Τhe measured area is divided into *N* > *M* grids. Assuming that the sound slowness in each grid follows a uniform distribution, Equation (2) can be converted into:(3)$$t_{j}=\overset{N}{\underset{i=1}{\sum }}f_{i}a_{ji},$$
where *f_i_* and *a_ji_* are the sound slowness value in the *i*th grid and the path length of the *j*th path in the *i*th grid, respectively. Define ***t*** = (*t*_1_, …, *t_M_*)*^T^*, ***A*** = (*a_ji_*), *j* = 1, …, *M*, *i* = 1, …,*N*, ***f*** = (*f*_1_, …, *f_N_*)*^T^*, consider all *M* acoustic paths and *N* grids, and obtain the AT measurement equation:(4)t=Af,
where ***f*** is the measured sound slowness signal of *N* × 1 dimension, ***A*** is the measurement matrix of *M* × *N* dimension, and ***t*** is the measurement result of *M* × 1 dimension. Using algorithms such as cross-correlation time delay estimation [10,18,20] the sound wave TOF can be measured on *M* sound wave paths and form TOF vector ***t***.

The TOF vector ***t*** is normalized with Equation (5) to obtain the normalized TOF vector. Variables $t_{H}$ and $t_{L}$ in Equation (5) represent TOF vectors under uniform temperature field *T* = *T*_H_ and uniform temperature field *T* = *T*_L_, respectively. Variables *T*_H_ and *T*_L_ are the highest and lowest temperatures that may occur in the preset test area. In this paper, *T*_L_ = 280 K and *T*_H_ = 400 K are taken for TOF normalization.
(5)$$t_{N}=\frac{t-t_{L}}{t_{H}-t_{L}},$$

The normalized TOF vector $t_{N}$ is inserted into the dictionary selector with the peak classifier as the core, and a sparse dictionary that best fits the measured temperature distribution can be selected from the preconstructed sparse dictionaries of *N* × *N* dimension. The sparse representation of sound slowness signal ***f*** is the sparse coefficient vector of *N* × 1 dimension ***θ*** in the sparce domain. The relationship between dictionary ***Ψ***, sound slowness signal ***f*** and the expression of ***f*** in sparse domain ***θ*** can be expressed as:(6)f=Ψθ,

The AT equivalent measurement equation shown in Equation (7) can be obtained by substituting Equation (6) into Equation (4), where ***U***=***A**Ψ*** is the equivalent measurement matrix and ***θ*** is the sparse equivalent measured signal.
(7)t=AΨθ=Uθ,

Calculating ***θ*** by ***t*** and ***U*** is a common sparse reconstruction problem. According to the compressed sensing theory [27,28], this problem can be reduced to the *l*_0_ norm optimization problem, as shown in equation (8):(8)$$\begin{array}{cc} \underset{\theta }{\min}\left\{||\theta |{|}_{0}\right\} & s.t.\;{\left\|t-U\theta \right\|}_{2}^{2}\leq r \end{array},$$
where *r*≥0 is the signal reconstruction error. Substituting the reconstructed sparse signal into Equation (6) yields the reconstructed sound slowness signal $\hat{f}$; then, the temperature distribution in the measured area can be obtained. Finally, the temperature distribution estimation $\hat{T}$ under the *N* grid can be reconstructed by using the relationship between sound velocity and temperature, shown in Equation (9):(9)$$\hat{T}={\left[1/\left(z\hat{f}\right)\right]}^{2},$$

Next, the four key steps of the MDL algorithm are described: sample set construction, dictionary construction, dictionary selector construction and sparse signal reconstruction.

### 2.2. Sample Set Construction

The sample set is used for dictionary learning, dictionary selector construction and algorithm performance testing. The measured area in this paper is a 1.3 m × 1.3 m square, with twelve acoustic transceivers distributed equidistantly around it. Sound waves are transmitted between transceivers to form *M* = 54 direct sound wave paths that effectively pass through the measured area. Figure 1 shows the direct acoustic path diagram:

In practice, the sample set can be constructed according to the characteristics of the temperature field, and then the dictionary and dictionary selector with superior adaptability to the signal can be trained. As a concrete example, this paper randomly adopts the model shown in Equation (10) [13,14] or (11) [17] to construct the temperature field set of six peak-types of temperature fields (average distribution(***P***_0_), unipeak (***P***_1_), double-peak (***P***_2_), three-peak (***P***_3_), four-peak (***P***_4_) and five-peak (***P***_5_)), where $T_{bg}$ is the background temperature of the temperature field, and peak is the temperature rise area on the background temperature in the temperature field, which is generated by the sum term contained in Equations (10) and (11). The number of “peaks” is the number of summation terms contained in the temperature field. The δ in Equations (10) and (11) is the number of “peaks” in the temperature field. In order to ensure the richness of the sample, δ takes an integer between 0 and 5; the remaining parameters are randomly selected in the following ranges:

*x_ξ_*∈[−0.4 × 1.3, 0.4 × 1.3], *y_ξ_*∈[−0.4 × 1.3, 0.4 × 1.3], *T*_bg_∈[290, 300], *T*_hp_*_ξ_*∈[60, 100]; when *δ* = 1, 2, 3, 4, 5, *sp_ξ_*_1_ take values in the range of [−25, −15], [−25, −15], [−50, −40], [−70, −60] and [−80, −70] in turn, *sp_ξ_*_2_ take values in the range of [30, 40], [30, 40], [100, 110], [120, 130] and [130, 140] in turn; when δ = 0, set $T_{hp\xi }$ = 0.
(10)$$T_{Exp}(x,y)=T_{bg}+\overset{\delta }{\underset{\xi =1}{\sum }}T_{hp\xi }\cdot e^{sp_{\xi 1}[{\left(x+x_{\xi }\right)}^{2}+{\left(y+y_{\xi }\right)}^{2}]},$$
(11)$$T_{Rp}(x,y)=T_{bg}+\sum_{\xi =1}^{\delta }\frac{T_{hp_{\xi }}}{sp_{\xi 2}[{\left(x+x_{\xi }\right)}^{2}+{\left(y+y_{\xi }\right)}^{2}]+1},$$

***P***_1_ ~ ***P***_4_ each contain 4000 samples, and ***P***_0_ and ***P***_5_ each contain 20 samples. ***P***_1_ ~ ***P***_4_ is randomly assigned in a ratio of 3:1 to form the training set ***P***_tr1_ ~ ***P***_tr4_ and the test set ***P***_te1_ ~ ***P***_te4_, each containing 3000 and 1000 samples, respectively. ***P***_0_ and ***P***_5_ are all used as tests. ***P***_tr1_ ~ ***P***_tr4_ is discretized with a grid of *N* = 15 × 15, and the relationship between temperature and sound slowness is used to convert them into the corresponding sound slowness training set ***F***_tr1_ ~ ***F***_tr4_, each containing 3000 samples, which is used for the training of each peak-type dictionary. A total of 4040 temperature field samples from ***P***_te1_ ~ ***P***_te4_, ***P***_0_ and ***P***_5_ are used to assess the temperature field reconstruction ability of the MDL algorithm.

To form a dictionary selector, 500 samples are randomly selected from each training set ***P***_tr1_ ~ ***P***_tr4_, and the corresponding normalized TOF vector is calculated to form the TOF training set of the peak classifier (called ***t***_Ntr1_ ~ ***t***_Ntr4_, each containing 500 samples). The normalized TOF vector corresponding to 4040 samples of ***P***_te1_ ~ ***P***_te4_, ***P***_0_ and ***P***_5_ is calculated to form the TOF test sets ***t***_Nte1_ ~ ***t***_Nte4_, each containing 1000 samples and ***t***_Nte0_, ***t***_Nte5_, each containing 20 samples, for performance testing of dictionary selectors.

### 2.3. Dictionary Construction

To increase the adaptability to signals of the learned dictionary, the dictionary construction in the MDL is performed by types, that is, each dictionary corresponds to a peak-type of temperature (sound slowness) distribution. If the signal type changes, the algorithm only needs to add or subtract the corresponding dictionary, which has no impact on other dictionaries.

This paper adopts the K-SVD dictionary learning algorithm [26] to the training dictionary. K-SVD is an adaptive dictionary design method based on training samples, and its objective function can be expressed as [26]:(12)$$\begin{array}{cc} \underset{\Psi ,\Theta }{\min}\left\{{\left\|F-\Psi \Theta \right\|}_{F}^{2}\right\} & s.t.||\theta_{i}|{|}_{0}\leq s,\forall i \end{array}=1,2,\cdots Q_{D},$$
where matrix ***F*** of *N* × *Q*_D_ dimension is the sound slowness distribution training sample set, each column is a sound slowness distribution sample, *Q*_D_ is the number of samples and *N* is the length of sound slowness samples, that is, the number of grids divided in the test area. Matrix ***Ψ*** of *N* × *K*_KSVD_ dimension is the object dictionary; in this paper, *K*_KSVD_ *= N*. Matrix ***Θ*** of *K*_KSVD_ × *Q*_D_ dimension is the sparse coding matrix and each of its column is a sparse vector of one sound slowness sample. Parameter *s* > 0 is the sparsity of the sparse coding vector $\theta_{i}$. Operator $||•|{|}_{F}$ is the Frobenius norm.

The learning dictionary with the K-SVD algorithm includes the following steps:1.Dictionary initialization and iteration number setting to *J* = 0. In this paper, the training sample set of sound slowness distribution ***F*** is decomposed by SVD, and the 1 ~ *K*_KSVD_ columns of the left singular matrix are used to form the initial dictionary;2.Sparse coding. For a given dictionary ***Ψ***, the OMP algorithm is hereby adopted to solve Equation (13), and to obtain the sparse vector ***θ****_I_* of each sample. To improve the sound slowness signal sparsity in dictionary sparse domain, the MDL stops the OMP iteration when the sparsity of ***θ****_i_* less than the ceiling of 1/4 times the AT system sound path number *M*, that is $||\theta_{i}|{|}_{0}<\left\lceil M/4\right\rceil$ (OMP iteration steps refer to Section 2.5);



(13)
$$\begin{array}{cc} \underset{\theta_{i}}{\min}\left\{||\theta_{i}|{|}_{0}\right\} & s.t.{\left\|f_{i}-\Psi \theta_{i}\right\|}_{2}^{2}\leq r,\forall i \end{array}=1,2,\cdots Q_{D},$$



3.Dictionary update. The sparse coding matrix ***Θ*** is fixed and the dictionary is updated column by column, using the SVD algorithm. The dictionary update is completed through *K*_KSVD_ iterations (For more detailed dictionary updating strategies, refer to [26]);

4.Assess whether the termination condition is met. If not, set *J* = *J* + 1 and return to step 2. Otherwise, terminate dictionary learning. The termination condition in this paper is: the ||***F***-***ΨΘ***||_2_ value of two adjacent times no longer decreases.

This paper used training set (***F***_tr1_, ***F***_tr2_, ***F***_tr3_, ***F***_tr4_) and the K-SVD algorithm to construct four dictionaries corresponding to four temperature (sound slowness) distribution types, namely single-peak, double-peak, three-peak and four-peak. As a contrast, 750 samples are randomly selected from ***F***_tr1_, ***F***_tr2_, ***F***_tr3_ and ***F***_tr4_ to form a mixed-peak-type sample training set, and a single dictionary covering four peak-type features is constructed using the K-SVD algorithm (parameters are set to *K*_KSVD_ = *N* = 15 × 15, *Q*_D_ = 3000).

### 2.4. Dictionary Selector Construction

The dictionary selector used the k-nearest neighbor (KNN) algorithm [29] to assess the peak-type of the current temperature field, and then selected a sparse dictionary that best matched the measured temperature field from a number of pre-constructed sparse dictionaries, based to the normalized TOF vector ***t***_N_, corresponding to the measured temperature field. The learning dictionary may weaken the matching degree of the unlearned signal types, while enhancing the matching degree of the learned signal types. Therefore, in addition to single-peak, double-peak, three-peak and four-peak dictionaries, the dictionary selector also contained a DFT dictionary, which has a good universality.

Principal components analysis (PCA) is used to extract features of the normalized TOF vector and improve the accuracy of peak pattern recognition of the KNN classifier. Therefore, the key to constructing the dictionary selector is to construct the PCA transformation matrix and the KNN peak classifier.

#### 2.4.1. PCA Transformation Matrix Construction

The PCA algorithm is based on the calculation of a group of new features from the original feature group, which are arranged in order of importance, from the highest to the lowest. The new features are linear combinations of the original features and are independent of each other. The mapping value of the original feature on the new feature is the new sample after feature extraction [30], and the mapping relationship is realized by the PCA transformation matrix. The TOF training set ***t***_Ntr1_ ~ ***t***_Ntr4_ is combined into a new sample set ***t***_Ntr_, which is also a matrix of *M* × *Q*_t_ dimension (*Q*_t_ = 2000), and the process of constructing a transformation matrix ***G*** is:5.The mean value of each row vector of the matrix ***t***_Ntr_ is subtracted from the mean value of the row vector to obtain the zero mean value matrix ***Y*** of the row vector; then, the covariance matrix ***C*** of the matrix ***Y*** is calculated:
(14)$$C=\frac{1}{N}YY^{T},$$

6.The eigenvalues and eigenvectors of matrix ***C*** are calculated, and the eigenvalues are arranged in descending order. The eigenvectors corresponding to the large eigenvalues of the previous *K*_PCA_ are arranged from top to bottom to form a dimensional matrix ***G***, which is the PCA transformation matrix. The value of *K*_PCA_ should make the contribution rate $\eta_{PCA}$ greater than the preset threshold. Equation (15) is the definition of contribution rate, where $\lambda_{j}$ is the *j*th largest eigenvalue. The preset threshold of *η*_PCA_ in this paper is 99.9%, the corresponding *K*_PCA_ = 50.



(15)
$$\eta_{PCA}=\left(\sum_{j=1}^{K_{PCA}}\lambda_{j}/\sum_{j=1}^{M}\lambda_{j}\right)\times 100\%,$$



By applying the PCA transformation matrix ***G***, features of the given *M* × 1-dimensional normalized TOF vector $t_{N}$ can be extracted, and the process is shown in Equation (16):(16)$$t_{G}=Gt_{N},$$
where ***t***_G_ is the *K*_PCA_ × 1-dimensional feature extraction vector from ***t***_N_. For TOF vector with *M* = 54 dimensions in this paper, the dimension after feature extraction is reduced to *K*_PCA_ = 50.

#### 2.4.2. KNN Classifier Construction

The KNN peak-type classifier is constructed by its training. KNN is a typical lazy learning algorithm [29]. Its training process only needs to record the samples in the training set, so the training time of the classifier is nearly zero. The KNN algorithm calculates the distance between training and test samples to identify the peak patterns. The previous *K*_KNN_ training samples that are close to the test samples are retained, the number of each type of label among them is counted and the type with the most labels is the classification result. In this analysis, the number was set to $K_{KNN}=5$. In the case that PCA feature extraction is also used, the KNN classifier will calculate the distance according to the samples after feature extraction.

To address the presence of unknown types, the most appropriate way is to add the corresponding sample set and train the corresponding dictionary and peak classifier. If unknown types have not been added to training set and have no contribution to the training dictionary and peak classifier, the KNN peak classifier will classify them as the closest known type, which might weaken the dictionary matching degree to the signal, and lead to a larger reconstruction error in the MDL algorithm. To avoid this situation, the peak-type classification reliability is hereby defined, according to Equation (17). When the reliability is less than 0.80, the classification result is considered untrustworthy, and the DFT dictionary with good universality is used instead:(17)$$\Delta_{KNN}=\Delta_{tr}/\Delta_{te},$$
where Δ*_te_* is the shortest distance (*l*_2_ norm) between the test and training samples and Δ*_tr_* is the largest distance (*l*_2_ norm) between all training samples in the same type.

#### 2.4.3. Performance Test of Dictionary Selector

As described in Section 2.4.1 and Section 2.4.2, a PCA-KNN dictionary selector (P-K dictionary selector) is constructed, using PCA for feature extraction and KNN as the classifier. Table 1 shows the performance test results of the dictionary selector. The test set consisted of four subsets, ***t***_Nte1_ ~ ***t***_Nte4_, ***t***_Nte0_ and ***t***_Nte5_, with a total of 4040 samples. To assess the effectiveness of feature extraction, Table 1 also shows the performance test results of the KNN dictionary selector (K dictionary selector) without feature extraction. The accuracy rate (AR) in Table 1 is defined as the number of the samples of dictionary that are correctly selected/the total number of samples in the test subset × 100%.

Table 1 shows that: (1) for peak-types that have been added to the training set, the AR of the P-K dictionary selector exceeded 99%, which is better than the K dictionary selector. This is because feature extraction improves the discrimination of samples; (2) For peak-types that have not been added to training set, that is, average distribution and five-peak, both K and P-K dictionary selectors selected the DFT dictionary correctly.

### 2.5. Sparse Signal Reconstruction

Trading off between algorithm time consumption and reconstruction accuracy, the OMP algorithm [27] is used to reconstruct the sparse signal. The OMP algorithm is a typical greedy algorithm. Compared with convex optimization, OMP takes less time [31,32], especially when the value of *N* is large. The steps of the OMP algorithm to solve Equation (8) are as follows:7.Initialization. Residual is set to ***r***_0_ = ***t***, index vector to ***Λ***_0_ = [], support matrix to ***V***_0_ = [] and iteration counter to *m* = 1;8.Column selection. The inner product between each column of matrix ***U*** and residual ***r***_m-1_ is calculated; then, the column vector that produces the maximum inner product to support matrix ***V***_m−1_ is added to form a new support matrix***V***_m_. The index that of the selected column vector is added in matrix ***U*** into index vector ***Λ***_m−1_ to form a new index vector ***Λ***_m_. Finally, the selected column is added in matrix ***U*** by zero vector;9.Residual update. Equation (18) is used to calculate ***θ***_m_ and the residual is updated by Equation (19):
(18)$$\theta_{m}={\left(V_{m}^{T}V_{m}\right)}^{-1}V_{m}^{T}t,$$
(19)$$r_{m}=t-V_{m}\theta_{m},$$

10.If the iteration termination conditions are met, iterations are stopped and the algorithm skips to step 5; otherwise, *m* = *m* + 1 is set and the algorithm returns to step 2, continuing the iterations;

11.A zero vector $\hat{\theta }$ of *N* dimension is constructed, and then $\hat{\theta }(\Lambda_{m})=\theta_{m}$ is set, which is the sparse signal reconstructed by OMP.

The same iteration termination condition as IMOMP is hereby adopted, that is, when $||r_{m}|{|}_{2}<\sqrt{M}\sigma$, iterations are stopped [18], where σ is standard deviation of noise in TOF vector ***t***. (The method to estimate σ in practical application is given in Section 3.2.)

## 3. Algorithm Verification

The reconstruction algorithm is implemented on a computer with Intel Core i8559 3.0 GHz CPU and 8 GB memory. In addition to the proposed MDL algorithm, five benchmark comparison algorithms are also introduced: single dictionary learning (SDL) algorithm, DFT Dictionary (DFTD) algorithm, CS-IMOMP algorithm, adopted in [18], LSA algorithm and SIRT algorithm. The only difference between the MDL, SDL and DFTD algorithms is the dictionary construction method. DFTD uses DFT as the dictionary, while SDL uses K-SVD algorithm and a mixed peak-type sample set to build a single dictionary. Both DFTD and SDL do not need a dictionary selector. CS-IMOMP also uses DFT as the dictionary, and its difference with DFTD is lies in the fact that it improves the OMP algorithm with variable column selection strategy. That is, multiple column selection is adopted in the initial stage of the iteration. After the change conditions are met, the single column selection strategy is adopted to ensure the “correctness” of column selection, which can reduce the reconstruction time of the temperature field to some extent [18]. The parameters in CS-IMOMP are *α* = 0.6, *p* = 3, and the column selection strategy change condition is ${\left||r_{m}|\right|}_{2}<\sqrt{M}\sigma$. The reconstruction formula of LSA and SIRT for Equation (4) is shown in Equations (20) [2] and (21) [12], respectively, and the meaning of each symbol is the same as Equations (3) and (4). Relaxation factor *α*_SIRT_ in Equation (21) is 0.3, and *k* is the iteration count of SIRT. When the conditions shown in Equation (22) are met, SIRT stops iterations.
(20)$$f={\left(A^{T}A\right)}^{-1}A^{T}t,$$
(21)$$f_{i}^{\left(k+1\right)}=f_{i}^{\left(k\right)}+\frac{\alpha_{SIRT}}{\sum_{j=1}^{M}a_{ji}}\overset{M}{\underset{j=1}{\sum }}\frac{a_{ji}\left(t_{j}-\sum_{i=1}^{N}a_{ji}f_{i}^{\left(k\right)}\right)}{\sum_{i=1}^{N}a_{ji}},$$
(22)$$\frac{{\left\|f^{\left(k+1\right)}-f^{\left(k\right)}\right\|}_{2}}{{\left\|f^{\left(k\right)}\right\|}_{2}}<1\times 10^{-4},$$

The measured area is divided into *N* = 15 × 15 = 225 grids for reconstruction algorithms MDL, SDL, DFTD and CS-IMOMP. The commonly used grid division, *N* = 5 × 5, is used for reconstruction algorithms LSA and SIRT, to meet the requirements that *N* < *M* and to avoid the seriously ill-conditioned inverse problem. The reconstruction results of the five algorithms are further described by cubic spline interpolation. In this paper, the root mean square error *E*_rms_ [4] is used to evaluate the reconstruction quality of the algorithm, which is defined as:(23)$$E_{rms}=\frac{1}{T_{mean}}\sqrt{\frac{1}{N_{f}}\overset{N_{f}}{\underset{i=1}{\sum }}{\left[T(i)-\hat{T}(i)\right]}^{2}}\times 100\%,$$
where *T*(*i*) and $\hat{T}(i)$ represent the temperature value on the *i*th pixel in the model temperature field and reconstruction temperature field, respectively; *T*_mean_ represents the average temperature of the model temperature field; *N*_f_ represents the number of refined pixels, in this paper, *N*_f_ = 37 × 37 = 1369.

### 3.1. Algorithm Verification Based on Simulation Data

The reconstruction performance of six temperature field reconstruction algorithms is tested by using the temperature field test sets (***P***_te1_ ~ ***P***_te4_) comprising known peak-types, and the temperature field test sets (***P***_0_ and ***P***_5_) comprising unknown peak-types. Table 2 shows the average value (${\bar{E}}_{rms}$) of six algorithms to reconstruct the temperature field test set. Each ${\bar{E}}_{rms}$ is the average of all sample $E_{rms}$ errors in the same sample set. TOF without noise is used to reconstruct temperature field.

The error results show that:12.Compared with the classical LSA and SIRT algorithms, the MDL, SDL, DFTD and CS-IMOMP produced lower reconstruction errors. This is because these algorithms divide the measured area into smaller grids, so the error caused by the assumption that the temperature distribution in the grid is uniform is smaller. Because CS-IMOMP uses the same dictionary as DFTD, their only difference is the column selection strategy of OMP iteration, so their reconstruction error is almost the same;13.When the peak-type of the test sample is the peak-type included in the training set, compared with SDL, especially in DFTD and CS-IMOMP, the MDL has obvious advantages in reconstruction error. Reasons for this include the dictionary construction by the learning method is better matched with the signal, the MDL adopts a multi-dictionary strategy, and each dictionary has better matching ability and sparse effect for its sound slowness signal of corresponding peak-type, while SDL uses a single dictionary to take account of multiple peak-types. While this improves its universality, its pertinence to specific peak-types will decline, resulting in its reconstruction error being generally slightly higher than the MDL, but still lower than DFTD and CS-IMOMP.14.When the peak-type of the test sample is absent from the training set, the P-K dictionary selector in the MDL algorithm selects the DFT dictionary, so the same reconstruction error as in DFTD is obtained. However, the reconstruction error of SDL is higher than that of DFTD, because the universality of the learned dictionary is lower than that of DFT dictionary, and the matching degree to samples that do not appear in the training set is also lower.

In practical application, noise in the TOF is unavoidable. Gaussian white noise with a mean value of zero and standard deviation of Δ*σ* = 6 × 10^−6^ is added to the TOF data to simulate the noise contained in the TOF vector during the actual measurement. Considering the randomness of the noise, 40 groups of noise for each TOF vector are generated, and 40 versions of TOF vectors that contain different noise are correspondingly constructed to simulate 40 times TOF measurements. The reconstruction error of the six reconstruction algorithms for any temperature field sample is the average of 40 times reconstruction errors, expressed by ${\bar{E}}_{rms}^{40}$. Table 3 shows the average reconstruction error (${{\bar{E}}^{\prime }}_{rms}$) of the six reconstruction algorithms for all temperature field samples in the test set, that is, ${{\bar{E}}^{\prime }}_{rms}$ is the average value of each ${\bar{E}}_{rms}^{40}$ in the same test set. The error results show that:

Comparing Table 2 and Table 3, it can be seen that:
15.For non-uniform temperature field, LSA error increases by 10.9% (***P***_te2_) and SIRT error increases by 9.6% (***P***_te2_) after adding noise. All are larger than the error growth based on sparse reconstruction algorithm, that is, algorithms based on sparse reconstruction are more robust to noise. For uniform temperature field, the sparse reconstruction algorithm only completes the reconstruction through one iteration, so it can be considered that it is not affected by noise;16.After adding noise to TOF, the error of the method based on sparse reconstruction is lower than that of the classical algorithm. On the one hand, the sparse reconstruction method has higher resolution, and its reconstruction error itself is smaller than that of the classical algorithm. On the other hand, the method based on sparse reconstruction uses the noise-related OMP iteration termination condition, so that it can stop the iteration in time when the effective data is close to the noise level;17.By comparing Table 2 and Table 3, we can see that the reconstruction error of MDL and SDL algorithms is still lower than that of DFTD and CS-IMOMP, even if TOF is noisy. This is mainly because the learned dictionary has better adaptability to sound slowness signal than the general DFT dictionary.

One temperature field sample is randomly selected in each sub test set (***P***_te1_ ~ ***P***_te4,_
***P***_0_, ***P***_5_), and one reconstruction image is also randomly selected from 40 images with noise of each algorithm. The reconstruction image is shown in Figure 2, in which the module temperature field is also given. Figure 2 shows that, compared with the classical LSA and SIRT algorithms, the reconstructed images of dictionary-based algorithms, that is MDL, SDL, DFTD and CS-IMOMP, are closer to the model temperature field. In addition, among the four dictionary-based reconstruction algorithms, the image that is produced by the MDL is the optimal.

It can also be seen from Figure 2 that, unlike other temperature fields, the peak boundary in 2-peak is fuzzy. This is because it is generated by Equation (11). Other temperature fields in Figure 2 are generated by equation (10). Whether it is generated by Equation (10) or Equation (11), it is randomly selected when generating the temperature field sample set. Figure 2 shows that no matter whether the temperature field is generated by Equation (10) or Equation (11), the reconstruction result of the MDL is the optimal.

Table 4 shows the reconstruction time comparison of the six reconstruction algorithms. The reconstruction time for each of them in Table 3 is the average value of the six model temperature fields reconstructed in Figure 2.

The results in Table 4 show that the reconstruction time of LSA is the shortest and that of DFTD is the longest. However, the reconstruction time of all algorithms is less than 4 ms, showing their good real-time performance. In a measurement cycle, the acoustic transceiver makes sound in turn, and other transceivers receive the acoustic wave. The reconstruction time needed for the four algorithms is far less than the measurement period. Among the four dictionary-based algorithms, the MDL has the shortest reconstruction time. The different reconstruction times of the four dictionary-based algorithms stem from the difference in the number of OMP iterations when reconstructing sparse signals. The average number of OMP iterations for the MDL, SDL, DFTD and CS-IMOMP is 15, 17, 23 and 12, respectively, for the reconstruction of the six model temperature fields in Figure 2. The number of iterations of the OMP algorithm also shows that the MDL dictionary has the strongest sparse ability for sound slowness signals, followed by the SDL dictionary, while the DFTD dictionary is the weakest. As the number of OMP iterations increases, the support set becomes larger, and the least squares estimation in OMP is prolonged. Therefore, the time consumption of DFTD is significantly greater than that of MDL and SDL. Moreover, the reconstruction time of CS-IMOMP is shorter than that of DFTD. This is because the multi-column selection strategy of CS-IMOMP enables it to complete sparse reconstruction more quickly with the same dictionary [18]. However, due to the matching degree of DFT dictionary to signals and the time required from the variable column selection strategy, the reconstruction time of CS-IMOMP is still longer than that of MDL and SDL, which also verifies that the dictionary built based on the learning method can improve sparse reconstruction efficiency.

### 3.2. Algorithm Verification Based on Measured Data

In this paper, the above-mentioned algorithms are tested on a self-designed AT temperature field reconstruction system. Figure 3 shows the hardware composition of the AT temperature field reconstruction system. The loudspeaker (TC9FD-19-08) diameter is 89.5 mm, while its length is 43.5 mm. Moreover, the microphone (CHZ 221) diameter is 7 mm, with a length of 61 mm, including the preamplifier (YG-221). The microphone and loudspeaker are connected through a metal bracket to form a sound wave transceiver, and the positions of the microphone and loudspeaker are shown in Figure 4, where S represents loudspeaker and C represents microphone. When setting the sound transceiver, the microphone is arranged in the transceiver position shown in Figure 1, and all sound transceivers are at the same height, that is, the height of the temperature measurement plane. Next, surrounded by 12 microphones, a square temperature measurement plane with a side length of 1.3 m is formed.

The audio signal is output from the sound card and is distributed to speakers 1 ~ 12 sequentially, using the bespoke power amplifier and audio distributor in a predetermined order. At any time, only one speaker sends the sound waves. In addition, the heat source used in the experiment is continuous and works with a fixed power. After a certain period of time, the temperature change in the measurement area usually tends to slow down and, eventually, the temperature value becomes stable. Therefore, in experiments, the temperature reconstruction is started after the temperature in the measurement area is stable. A complete TOF measurement cycle is about 6 s, which essentially ensures that the temperature is relatively stable in the TOF measurement cycle.

The output signals of the 12 microphones are amplified by 12 preamplifiers (YG-221) and input to the signal conditioner (PM16-12). The conditioned signal is then transmitted to the computer through the data collector, which comprises PCI 6123 data acquisition cards. Each PCI 6123 card has eight synchronous sampling analog input channels, with a maximum sampling rate of 500 kS/s, which can reduce the maximum value of quantization error of TOF to 2μs. When the loudspeaker sends sound waves, the data collector can synchronously sample the amplified output signals of the 12 microphones and input them into the computer. Based on these collected data, TOF data along the effective acoustic path can be calculated using the time delay estimation program, and then the reconstructed temperature distribution can be obtained from the temperature distribution reconstruction program.

This paper uses the cross-correlation (CC) method to estimate the TOF. When two acoustic signals are received by the microphone at the transmitter and receiver, calculate their cross-correlation functions and find the peak values of these cross-correlation functions. It should be noted that the time corresponding to the peak value is TOF. Since the noise in the temperature measurement environment is often random and irrelevant to the audio signal in the experiment, CC method can effectively suppress the noise. In order to further suppress the noise interference, it is necessary to select an appropriate audio signal.

The audio signal used in this paper is the maximum length sequence (MLS), which is a pseudo-random binary sequence. An important property of any MLS is that its auto-correlation function increases and approaches the unit pulse function as the MLS length increases. Moreover, MLS produces a lower secondary maxima in the cross-correlation curve than the chirp sequence, implying a better resistance to the noise arising from the misinterpretation of “true maximum” corresponding to the correct time delay, that is to say, while both chirp sequence and MLS can be used in boilers [33] and grain bulk [34], MLS has a more consistent time-delay estimation than the chirp sequence [33,34].

The experiment reported in this paper is carried out indoors, so there is a certain amount of acoustic reflection. Since the reflected wave and audio signal are of the same origin, a peak value corresponding to the reflected signal will also be formed in the cross-correlation function. When this peak value is close to the peak value of direct sound wave, the TOF data calculated by the CC method may be unstable. To overcome this, the method given in reference [20] is used herein to suppress the influence of reflected waves. First, calculate the ratio of the two peaks; then, judge whether the ratio reaches the given threshold value, so as to judge the position of direct acoustic peak (See literature [20] for details about the methods).

Because of the separation between loudspeaker and microphone, for any pair of transceivers receiving and transmitting the sound waves, if loudspeaker S and microphone C in the same transceiver and microphone D in other transceiver to receive the sound waves are not in the same line, the theoretical sound wave path length *l* and the actual length *l*_1_ will inevitably be different.

As shown in Figure 4, a theoretical sound path length between the two transceivers is calculated as the distance between microphones C and D, recorded as *l*. However, the direct sound path is on the line from S to D. Since the sound wave emitted by S propagates approximately as a spherical wave, the actual sound line length is C_1_ to D, which is recorded as *l*_1_. The C_1_ is often different than C, which leads to errors in the TOF measurement. This paper uses the method in [35] to correct this error. Under a certain uniform temperature distribution, calculate the theoretical value of TOF $t_{sys}$ on each acoustic line, and measure the actual value of TOF $t_{act}$; then, calculate the TOF correction coefficient $\mu =t_{sys}/t_{act}$ on each sound path. Here, the influence of temperature change on the correction coefficient can be ignored, and the correction coefficient obtained at a certain temperature can be used for other temperature values as well. Therefore, in actual TOF measurements, the TOF error can be corrected by multiplying the measured TOF by the corresponding correction coefficient.

For algorithms, the dictionary and peak classifier used in the algorithms are all trained in Section 3.1. The method described in [18] is hereby used to estimate the parameters *σ* required by the MDL, SDL, DFTD and CS-IMOMP. First, the sampling frequency of the acoustic signal by the data acquisition card is 500 k Hz, so the maximum value of quantization error of TOF is $\sigma_{s}=2\;\mu s$. Then, 40 consecutive TOF measurements are performed under the average temperature field of 300.8 K, and the average standard deviation of TOF multiplied by correction coefficient of all sound paths is calculated as $\sigma_{m}=1.1\;\mu s$. Finally, the larger value of $\sigma_{s}$ and $\sigma_{m}$ is considered as the estimated value of noise standard deviation *σ* contained in TOF, that is $\sigma =2\times 10^{-6}$.

In this paper, two types of temperature fields are designed for reconstruction experiments, namely non-uniform and uniform temperature fields. The non-uniform temperature field is a single-peak and double-peak temperature field, produced with an electric furnace as the heat source. The heat source layout is shown in Figure 5, marked as T1, T2 and T3, respectively. The uniform temperature field is a 300.8 K indoor environment without heat source, which is recorded as T4. All temperature fields are reconstructed by the device shown in Figure 5, which is a square the same size of that in the simulation, that is 1.3 m × 1.3 m.

Figure 6 shows the images of T1 ~ T4 reconstructed by the six algorithms. An 80PK-1 thermocouple and digital thermometer Fluke 54 II are additionally employed in this study to obtain the temperature measurement value at the location of the hot spot (i.e., the intersection of the yellow line) on the test level, as a further way to compare the reconstruction results among algorithms. In T1, the thermocouple position is (0,0); in T2, the thermocouple position is (−0.25, 0.25); and in T3, the two thermocouple positions are (−0.25, 0.25) and (0.30, −0.30), respectively, named T3-1 and T3-2. Figure 7 shows the temperature comparisons measured by the thermocouple, with the reconstruction temperature of different algorithms. The error *e* (%) of the reconstructed temperature relative to the measured temperature is also given in Figure 7b, that is *e* = [(reconstructed value—measured value)/measured value] × 100%. The hot spot (peak) temperature for the non-uniform temperature field is given in Figure 7a, while for the uniform temperature field T4, the maximum temperature *T*_max_ and minimum temperature *T*_min_ are given in the Figure. The last group of Figure 7b shows ${\left|e\right|}_{sum}$, that is the sum of each absolute value of *e* of each algorithm.

It can be seen from Figure 6 and Figure 7 that:18.Although the six reconstruction algorithms can successfully reconstruct the basic characteristics of the measured temperature field, there are noticeable differences in the shape, color and artifacts of the hot spot (peak) in the reconstructed image. The reconstruction result of dictionary-based algorithms is better than LSA and SIRT. Among dictionary-based algorithms, the MDL is the best, SDL reconstruction results are generally better than DFTD and CS-IMOMP, and DFTD and CS-IMOMP reconstruction results are almost identical;19.For the non-uniform temperature fields T1~T3, the area of artifacts in the MDL reconstruction image is small and not noticeable, and the error of the hot spot reconstruction temperature relative to the measured temperature is the smallest among the six algorithms. Therefore, the MDL provides the optimum reconstruction effect. The artifact area refers to the area where the reconstruction temperature is lower than the measured value;20.The MDL peak classification confidence is lower than the given threshold for uniform temperature field T4. Because the dictionary selector uses a DFT dictionary, the MDL and DFTD algorithms reconstruct the image with the same error. SDL uses a learning dictionary, whose reconstruction error is higher than that of DFTD because of its lower generalization ability.

Table 5 shows the average time consumption of the six algorithms to reconstruct T1~T4 in the experiment. Similar to data in simulation (Table 4), LSA takes the shortest time, followed by SIRT. Among all algorithms based on sparse reconstruction, the MDL takes the shortest time, SDL takes the second place, and DFTD takes the longest time. Because the temperature distribution of T1~T4 is simpler than that of 3-, 4- and 5-peaks, and requires fewer iterations, the sparse reconstruction algorithm takes less time in experiment than in numerical simulation.

## 4. Conclusions

This paper presents an acoustic CT (AT) temperature field reconstruction algorithm based on multi-dictionary learning (MDL) which consists of two phases, namely, offline training and online rebuilding. The K-SVD dictionary learning algorithm and the sample set of acoustic slowness signals constructed by distinct peak-types are used to build corresponding dictionaries for different peak temperature fields in the offline training phase. A dictionary selector, based on primary component analysis (PCA) for feature extraction of time of flight, (TOF) signals and a k-nearest neighbor (KNN) peak classifier and P-K dictionary selector, is constructed. In the online reconstruction phase, the dictionary selector is used to select the optimal matching dictionary peak from the pre-trained dictionary, according to TOF; then, the OMP algorithm is used to reconstruct the sparse representation of the sound slowness signal in the sparse domain of the selected dictionary and obtain the sound slowness and temperature distribution. The simulation and actual temperature field reconstruction experimental results show that the MDL algorithm has lower reconstruction error and is closer to the true distribution than the classical LSA and SIRT algorithms. The MDL has lower reconstruction error and shorter time requirements than compressed sensing and improved orthogonal matching pursuit (CS-IMOMP). The results demonstrate that the proposed MDL method has better matching degree than the analytic method, and that building a dictionary with higher matching degree to sound slowness is more effective than simply optimizing the OMP algorithm column selection strategy to improve the efficiency of temperature field reconstruction. The MDL may have application limitations for temperature distributions with insignificant characteristics. Because the peak classifier is difficult to classify at this time, the MDL may choose the wrong dictionary. To overcome this difficulty, when the peak recognition reliability is less than a certain threshold, the MDL uses a more versatile DFT dictionary.

The MDL algorithm constructs corresponding dictionaries for the sound slowness signals of different peak temperature fields and uses dictionary selection to achieve the combination of multiple dictionaries. Therefore, the MDL algorithm has the following advantages: (1) Dictionary specialty and coverage are considered simultaneously, so the sparse effect is good; (2) Adding or deleting peak shapes does not affect the existing dictionary. By adding or deleting the corresponding TOF samples in the dictionary selector and training the corresponding dictionary for the new peak shapes, the MDL can adapt to temperature fields with different characteristics. In fact, the MDL algorithm provides an open idea for temperature field sparse reconstruction. Its peak classifier and sparse signal reconstruction can be optimized as needed. For example, a peak-type classifier based on artificial neural network is constructed to improve the recognition rate of temperature field type, and then the correct dictionary can be selected. Using a convex optimization algorithm (such as FISTA) for sparse reconstruction can obtain higher sparse reconstruction accuracy than that of OMP algorithm, but it will consume more time. Therefore, the MDL algorithm has great practical value.

## Figures and Tables

**Figure 1 sensors-23-00208-f001:**
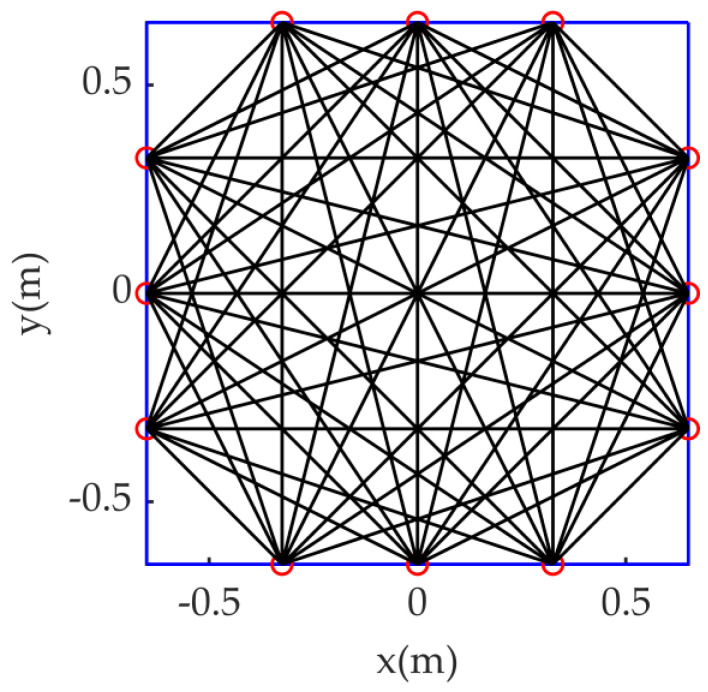
Schematic diagram of the direct acoustic wave path.

**Figure 2 sensors-23-00208-f002:**
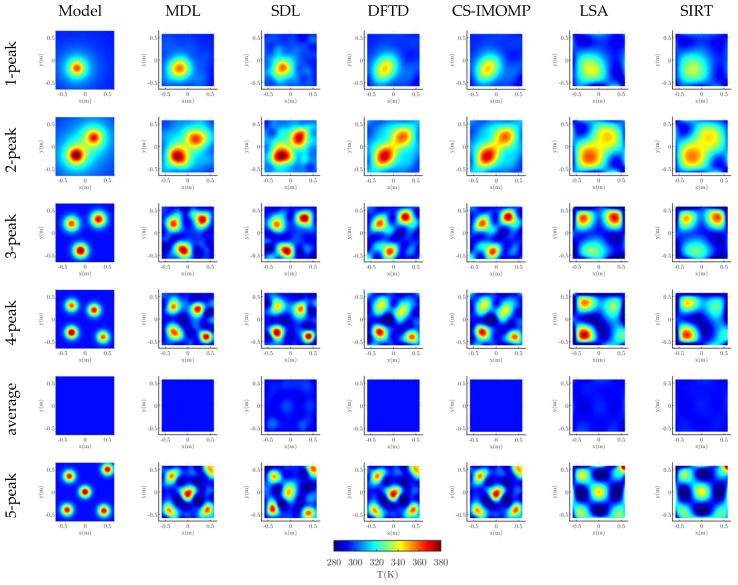
Model temperature fields and reconstructed temperature fields.

**Figure 3 sensors-23-00208-f003:**
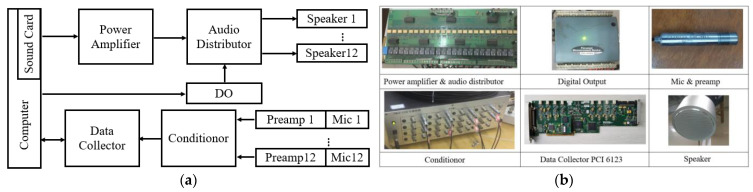
System structure block diagram: (**a**) System structure block diagram; (**b**) System structure block diagram.

**Figure 4 sensors-23-00208-f004:**
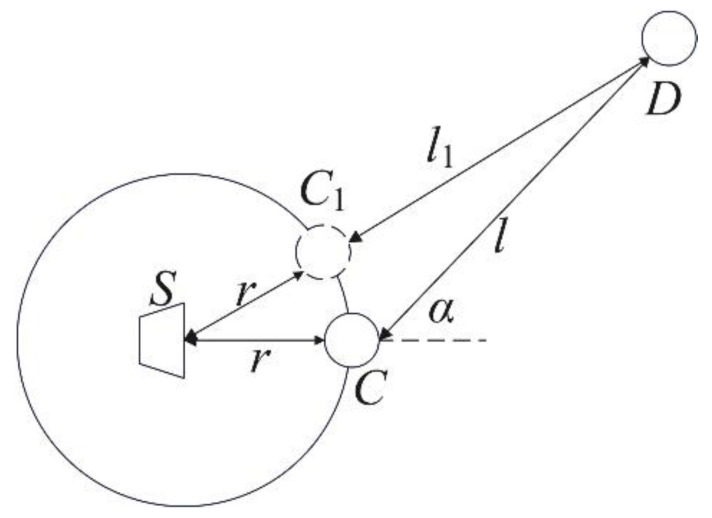
Theoretical sound path length *l* and actual sound path length *l*_1_.

**Figure 5 sensors-23-00208-f005:**
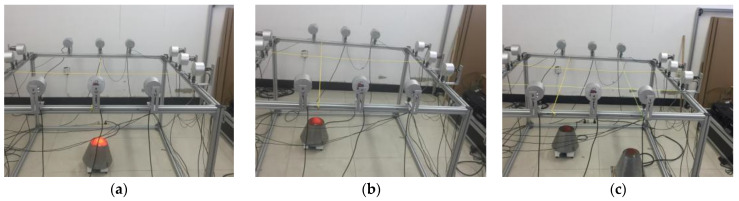
Heat source arrangement of non-uniform temperature fields: (**a**) Temperature field T1; (**b**) Temperature field T2; (**c**) Temperature field T3.

**Figure 6 sensors-23-00208-f006:**
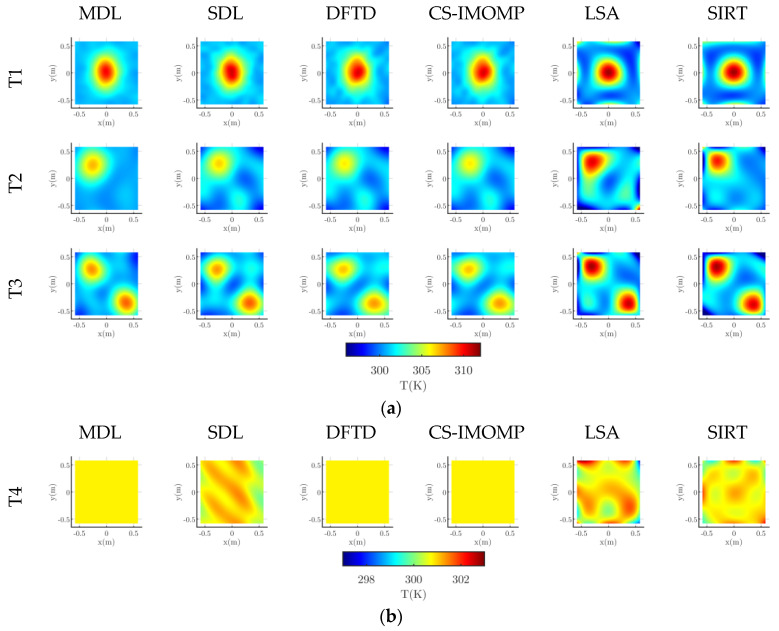
Reconstructed images of the actual temperature fields: (**a**) Reconstructed images of the T1-T3; (**b**) Reconstructed images of the T4.

**Figure 7 sensors-23-00208-f007:**
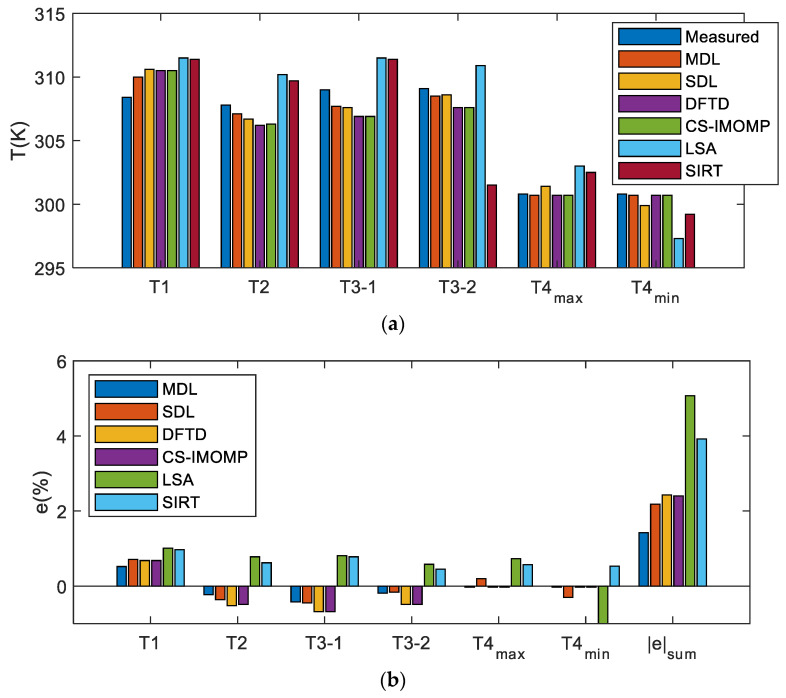
Experiment data: (**a**) Measured value and reconstructed value; (**b**) error of the reconstructed value relative to the measured value.

**Table 1 sensors-23-00208-t001:** Accuracy rate of dictionary selection.

TOF Test Set	Correct Dictionary Type	AR (%) of K Dictionary Selector	AR (%) of P–K Dictionary Selector
* **t** * _Nte1_	single-peak	100%	100%
* **t** * _Nte2_	double-peak	98.20%	99.96%
* **t** * _Nte3_	three-peak	94.80%	99.52%
* **t** * _Nte4_	four-peak	87.80%	99.20%
* **t** * _Nte0_ * **t** * _Nte5_	DFT dictionary	100%	100%

**Table 2 sensors-23-00208-t002:** Comparison of average of reconstruction errors ${\bar{E}}_{rms}$ (%) reconstructed by TOF without noise.

Test Set	MDL	SDL	DFTD	CS-IMOMP	LSA	SIRT
* **P** * _te1_	0.94	1.08	1.11	1.11	1.75	1.49
* **P** * _te2_	1.04	1.34	1.44	1.44	1.92	1.77
* **P** * _te3_	1.56	1.95	2.16	2.16	2.93	2.91
* **P** * _te4_	1.96	2.34	2.59	2.59	3.61	3.58
* **P** * _0_	0.01	0.01	0.01	0.01	0.04	0.04
* **P** * _5_	3.16	3.28	3.16	3.16	4.76	4.82

**Table 3 sensors-23-00208-t003:** Comparison of average of reconstruction errors ${{\bar{E}}^{\prime }}_{rms}$ (%) reconstructed by TOF with noise.

Test Set	MDL	SDL	DFTD	CS-IMOMP	LSA	SIRT
* **P** * _te1_	1.02	1.18	1.22	1.22	1.92	1.63
* **P** * _te2_	1.12	1.45	1.54	1.53	2.13	1.94
* **P** * _te3_	1.63	2.01	2.23	2.23	3.10	3.05
* **P** * _te4_	2.14	2.43	2.67	2.67	3.80	3.73
* **P** * _0_	0.01	0.31	0.01	0.01	0.66	0.39
* **P** * _5_	3.25	3.48	3.25	3.25	4.96	5.00

**Table 4 sensors-23-00208-t004:** Reconstruction time (ms) in simulation.

MDL	SDL	DFTD	CS-IMOMP	LSA	SIRT
1.16	1.58	3.66	2.29	0.04	0.64

**Table 5 sensors-23-00208-t005:** Reconstruction time (ms) in experiment.

MDL	SDL	DFTD	CS-IMOMP	LSA	SIRT
1.09	1.42	3.58	2.18	0.04	0.62

## Data Availability

Not applicable.

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
