# Peer review of "Temperature Field Reconstruction Method for Acoustic Tomography Based on Multi-Dictionary Learning"

_sensors, 2022, doi:10.3390/s23010208_

Round 1

Reviewer 1 Report

1. Why some of the paragraphs have background with green color?

2. The algorithm based on MDL for AT should have the application limits, such as the temperature distribution is sparse.

3. In introduction, the description of other related algorithms based on sparsity is lack.

4. The unit of abscissa variable is missing in Figs. 2 and 5.

5. In Figure 2, the model of 2-peak is maybe wrong.

6. The size of the sensor in Figure 4 should be given. Is the same as the size used in simulation?

7. The reconstructed results with noise-containment TOFs in simulation should be discussed.

8. The precision of OMP algorithm is not high although the speed is fast. Why the authors do not consider convex optimization algorithm?

Round 2

Reviewer 1 Report

the authors  have made good modifications. The size of experimental device in  Fig. 5 should be supplemented, not the size of the sensors in Fig. 3.
